# Prompt Tuning Vision Language Models with Margin Regularizer for Few-Shot Learning under Distribution Shifts

**Debarshi Brahma**                                            *debarshib@iisc.ac.in*
*Indian Institute of Science, Bangalore*

**Anuska Roy**                                                   *anuskoroy@gmail.com*
*Indian Institute of Science, Bangalore*

**Soma Biswas**                                                *somabiswas@iisc.ac.in*
*Indian Institute of Science, Bangalore*

**Reviewed on OpenReview:** *https://openreview.net/forum?id=ZnWqtPhHM7*

## Abstract

Recently, Vision-Language foundation models like CLIP and ALIGN, which are pre-trained on large-scale data have shown remarkable zero-shot generalization to diverse datasets with different classes and even domains. In this work, we take a step further and analyze whether these models can be adapted to target datasets having very different distributions and classes compared to what these models have been trained on, using only a few labeled examples from the target dataset. In such scenarios, finetuning large pretrained models is challenging due to problems of overfitting as well as loss of generalization, and has not been well explored in prior literature. Since, the pre-training data of such models are unavailable, it is difficult to comprehend the performance on various downstream datasets. First, we try to answer the question: *Given a target dataset with a few labelled examples, can we estimate whether further fine-tuning can enhance the performance compared to zero-shot evaluation?* by analyzing the common vision-language embedding space. Based on the analysis, we propose a novel prompt-tuning method, *PromptMargin* for adapting such large-scale VLMs directly on the few target samples. PromptMargin effectively tunes the text as well as visual prompts for this task, and has two main modules: 1) Firstly, we use a selective augmentation strategy to complement the few training samples in each task; 2) Additionally, to ensure robust training in the presence of unfamiliar class names, we increase the inter-class margin for improved class discrimination using a novel Multimodal Margin Regularizer. Extensive experiments and analysis across fifteen target benchmark datasets, with varying degrees of distribution shifts from natural images, shows the effectiveness of the proposed framework over the existing state-of-the-art approaches applied to this setting. Code: `https://github.com/debarshigit/PromptMargin`.

## 1 Introduction

With the rapid advancement of deep learning models, it is now possible to achieve very high performance for tasks like classification, etc., where large amounts of training samples can be collected and annotated. However, in real-world scenarios, the difficulty in curating huge amounts of labelled data has led to research in Few-Shot Learning (FSL), where a model trained on a large dataset can be transferred to a downstream task having few labelled samples from unseen categories. Furthermore, the target distribution can be very different from the source distribution, making the problem even more challenging. Recently, vision-language foundation models like CLIP have shown remarkable generalization capabilities in the zero-shot scenarios (Radford et al., 2021). Efficient finetuning techniques like Prompt learning (Zhou et al., 2022b;a) have

been successful in generalizing these models to new classes or new domains separately, but adapting such multimodal models to few samples having both novel categories and domains simultaneously is relatively less explored. Foundation models like CLIP have been pretrained on large web-scale data, which is not public. Hence, it is unclear as to which classes and domains does this training data encompass, and what is really in-distribution (ID) and out-of-distribution (OOD) for such models. In this work, we first explore whether we can estimate CLIP's performance on a given target dataset from the text and image embeddings in the joint representation space, even before finetuning the model using the data. Based on this observation, we aim to enhance the performance of CLIP on such datasets using prompt learning. For this, we propose a novel prompt-learning framework, termed **PromptMargin** in a completely source-free setting. This implies that, unlike standard frameworks of few-shot learning, where the pre-trained model is meta-trained on a source domain (e.g., ImageNet (Russakovsky et al., 2015)) before it is fine-tuned on the downstream dataset, we utilize a more practical setting of directly adapting the original pre-trained CLIP model on the few target samples. PromptMargin has two main modules, namely (i) Selective Augmentation and (ii) Multimodal Margin Regularizer, to handle the different challenges in this setting. To address the challenge of availability of few samples from the target data, we use a selective augmentation strategy to increase the number of training samples. Prompt tuning generally relies on the class names of the new categories, which may not be available for the target dataset or the names may not be meaningful to the CLIP model (e.g., names of rare diseases, etc. which may not have been seen during training). In order to learn discriminative classifiers even in such scenarios, we propose a novel Multimodal Margin Regularizer (MMReg), which enforces a consistent separation between the class-wise embedding vectors in the joint vision-language representation space. Extensive experiments on benchmark datasets, namely BSCDFSL (Guo et al., 2020), and Meta-dataset (Triantafillou et al., 2019), show that the proposed framework performs favourably compared to the state-of-the-art, even though it is fine-tuned in a completely source-free manner. Our contributions can be summarized as follows:

1. We first empirically explore CLIP's zero-shot performance on a few target datasets with limited labels, by investigating the image and text feature distances in the multimodal representation space. This provides insights into when CLIP's performance is sufficient for some downstream task, and how to enhance its performance in other cases.

2. We propose a novel prompt learning framework *PromptMargin* where we adapt CLIP directly to a few-shot setting with different classes as well as distribution shifts, without training it on some source dataset. To the best of our knowledge, this is the first work which addresses this task using vision-language models (VLMs).

3. Towards this goal, we introduce a novel Multimodal Margin Regularizer (MMReg), which jointly steers the image and text category embeddings to uniform separation, thereby improving the prompt-learning performance.

4. Our framework performs favourably over zero-shot, meta-training, and baseline prompt-learning methods across fifteen benchmark datasets in 1-shot and 5-shot settings.

Next, we discuss the related work in literature followed by the details of the proposed framework and experimental evaluation.

## 2 Related Works

Here, we briefly discuss the related work on prompt learning and OOD few-shot learning.

**Prompt Learning for Vision-Language Model (VLM):** The recent emergence of foundation VLM's like CLIP (Radford et al., 2021), and ALIGN (Jia et al., 2021), has changed the landscape of deep learning. These models are trained on abundant web-scale data, where they align the image-text representations in a contrastive manner, exhibiting remarkable zero-shot performance. Though such models generalize well to most cases, leveraging the knowledge learned by these models for downstream tasks is a challenging task. Recently, prompt learning has emerged as an effective choice for finetuning these

large-scale models to downstream tasks, where a few additional trainable parameters are added to the input branches. Prompt learning in VLM's was first explored by CoOp (Zhou et al., 2022b), where the handcrafted prefix of the text input was replaced by a few trainable parameters, to be finetuned for a classification task. CoCoOp (Zhou et al., 2022a) addressed the reduced generalization of CoOp in certain cases by conditioning the text prompts on the image embeddings. MaPLe (Khattak et al., 2023) first introduced the concept of multimodal prompt learning, where a coupling function was utilized to enable mutual synergy between the textual and visual prompts. This approach enabled joint training of both the prompts in the CLIP representation space, and demonstrated improved performance over unimodal prompting approaches. This state-of-the-art model serves as the baseline of our proposed framework, where we learn prompts in both the text and vision encoder branches.

**Few-Shot Learning:** Few-Shot Learning (FSL) aims to transfer a trained model to novel category data when a few number of samples are available from each class (Snell et al., 2017; Sun et al., 2019; Xu et al., 2021). Existing literature provides two approaches to this problem, namely meta-learning and transfer-learning. Meta-learning based approaches typically aim to simulate few-shot tasks on a source dataset, and then transfers this learner to the test domain tasks (Ravi & Larochelle, 2016; Finn et al., 2017). The conventional transfer learning approaches are explored by methods like BSCDFSL (Guo et al., 2020), BSR (Liu et al., 2020), and NSAE (Liang et al., 2021), where the models are first trained on a source dataset like ImageNet, before finetuning them on the target datasets. Recently, the prominence of foundation models saw the emergence of a variety of methods for adapting them to the few-shot learning task. For instance, FDAlign (Song et al., 2024) and Wise-FT (Wortsman et al., 2022), finetunes the entire CLIP model parameters on a source data using regularization techniques to avoid loss of rich representations of the original model. Wise-FT utilizes weight interpolations between the zero-shot and the fine-tuned models while training, to enhance the performance. FD-Align aims to maintain the spurious correlations intact before and after finetuning, by minimizing the divergence between the predictions of the two models. Training free approaches like Tip-Adapter (Zhang et al., 2021) constructs weights from a key-value cache model from the few-shot training set, to adapt the CLIP model without any backpropagation. An intermediate approach of parameter-efficient finetuning is adopted by CoOp (Zhou et al., 2022b), MaPLe (Khattak et al., 2023), which first trains the prompts on a source dataset before transferring them to the target dataset as discussed earlier.

*Though prompt learning based approaches have been utilized in VLMs for FSL, to the best of our knowledge, no work in literature addresses the additional significant distribution shift problem in this context. Inspired by these advances, our proposed PromptMargin aims to address this task using an effective regularization technique for prompt learning, in a completely source-free manner.*

## 3 Analyzing the CLIP Representation Space for target datasets

Foundation models like CLIP have been pretrained on a large corpus of web-scale data, which is not publicly known, and hence may span a wide variety of classes and domain data, ranging from standard academic datasets to specialized datasets. Recent papers like Udandarao et al. (2024) have studied the possible relationship of zero-shot performance of CLIP with the concept frequencies in pretraining datasets. A pertinent question in this scenario is, *Given a target dataset with a few labelled examples, can we estimate whether further fine-tuning can enhance the performance compared to zero-shot evaluation?* The zero-shot generalizability of the model depends upon both the distribution and semantic difference of the target dataset from the CLIP training dataset. Since the source dataset is unavailable and only few samples of the target dataset are provided, directly estimating distribution difference and also understanding whether the target classes are seen or not is not straightforward. Here, we propose a simple method to estimate the extent of the distribution shifts of some of the target datasets, by relating their inter-class mean image and text embedding distances in the CLIP representation space.

We consider some representative datasets from BSCDFSL (Guo et al., 2020) and MetaDataset (Triantafillou et al., 2019) benchmarks to illustrate this point. We estimate the distribution and semantic difference using the mean inter-class $L_2$ distances of both text and image features from the frozen zero-shot CLIP model. The class text features obtained by passing "A photo of [CLASS]" through the zero-shot model is denoted

Table 1: The mean inter-class text and image embedding distances along with the estimated combined (semantic and distribution) differences for some target datasets. "*" denotes that pseudo class names were used for that particular dataset, due to unavailability. We replace $m_T$ with a small value (0.1) for such cases.

| Dataset | EuroSAT | ISIC | Omniglot | Quickdraw | Plantae* | Traffic Signs* | MSCOCO* | Aircraft | mini-ImageNet |
|---|---|---|---|---|---|---|---|---|---|
| $m_T$ | 0.588 | 0.561 | 0.679 | 0.716 | 0.100 | 0.100 | 0.100 | 0.700 | 0.860 |
| $m_V$ | 0.627 | 0.582 | 0.420 | 0.520 | 0.727 | 0.583 | 1.010 | 0.770 | 1.010 |
| $diff(m_T, m_V)$ | 1.296 | 1.500 | 1.854 | 1.320 | 9.376 | 9.715 | 8.990 | 0.730 | 0.153 |
| ZS-CLIP (%) | 47.70 | 22.40 | 28.14 | 61.54 | 26.54 | 12.68 | 18.61 | 80.98 | 99.21 |
| MaPLe (%) | 75.46 | 31.96 | 77.82 | 72.54 | 55.34 | 56.45 | 53.09 | 79.76 | 99.17 |

by $\tilde{X}_{z_T}$ and the image feature prototypes are denoted by $\tilde{X}_{z_V}$. The inter-class mean distances $m_T$ and $m_V$ are computed as follows:

$$m_T = \frac{2}{C^2 - C} \sum_{i<j} \|\tilde{X}_{z_{T_i}} - \tilde{X}_{z_{T_j}}\|_2, \ \forall j \in \{2, 3, ..., C\} \tag{1}$$

$$m_V = \frac{2}{C^2 - C} \sum_{i<j} \|\tilde{X}_{z_{V_i}} - \tilde{X}_{z_{V_j}}\|_2, \ \forall j \in \{2, 3, ..., C\} \tag{2}$$

Here, $C$ is the total number of classes. The combined estimated distribution and semantic difference is approximated as follows:

$$diff_{\mathcal{D}_{target}}(m_T, m_V) = \left( \frac{1}{m_T} + \frac{1}{m_V} - 2 \right) \tag{3}$$

When the target dataset $D_{target}$ has significant semantic and distribution difference with the original training data, the CLIP model will not be able to distinguish between the image and text embeddings of the target dataset. Thus, the values of $m_T$ and $m_V$ will be smaller and the difference $diff_{\mathcal{D}_{target}}$ will be larger and vice-versa. Since, the extracted feature vectors are normalized between 0 and 1, we subtract 2 as an offset from $diff_{\mathcal{D}_{target}}$ to set its minimum value to zero, while preserving the relative differences.

Table 1 shows the difference along with the accuracy of zero-shot CLIP (ZS-CLIP) and MaPLe (Khattak et al., 2023), where both the image and text encoders were fine-tuned using the few available labeled target samples using prompt learning. We observe that for the first four datasets, $diff_{\mathcal{D}_{target}}$ is large, i.e. the classes are not well separated, and thus there is scope for improvement over zero-shot CLIP accuracy. This is consistent with the improvement obtained with MaPLe. Similarly, ZS-CLIP performs very poorly when placeholder classnames are used instead of original names due to their unavailability (Plantae, Traffic Signs and MSCOCO datasets), thus implying the possibility of significant performance improvement, as can be seen in MaPLe. On the other hand, for mini-ImageNet and Aircraft, since $diff_{\mathcal{D}_{target}}$ is low (between 0 and 1), the classes are already well separated in the latent space, and fine-tuning using few samples can adversely affect the model, thereby justifying the drop in accuracy of MaPLe. We also compute the Pearson correlation between the ZS-CLIP performance and the $diff_{\mathcal{D}_{target}}$ metric , which is $-0.713$, and shows a strong negative correlation. This conforms with the empirical analysis above, suggesting that an increase in the $diff_{\mathcal{D}_{target}}$ metric results in reduced zero-shot performance of CLIP.

This analysis illustrates that the relative separations between image and text features for the different classes in the CLIP representation space is crucial in explaining the zero-shot performance on the respective downstream datasets. We address this issue by introducing a simple and effective regularization framework for prompt tuning, where we guide the image and text features to separate out in the feature space. We now formally give the problem definition and describe our proposed approach in detail.

## 4 Problem Definition and Background

We address the problem of transferring a model trained on a large source dataset to a target domain containing very few labeled training examples and having significantly different data distribution. For this,

we consider the *N-way k-shot* episodic setting, where random tasks/episodes $\mathcal{T}$ are sampled, comprising of a support set $\mathcal{S}$, and a query set $\mathcal{Q}$. Both $\mathcal{S}$ and $\mathcal{Q}$ contains $N$ classes randomly selected from among all the novel categories of the target dataset. For the *N-way k-shot* setting, $k$ samples are drawn from each of these $N$ sampled classes to create the support set. Additionally, $q$ samples are also drawn from the same classes to create the query set. The support and query sets are given by $\mathcal{S} = \{(X_i, y_i)\}_{i=1}^{N \times k}$, $\mathcal{Q} = \{(X_i, y_i)\}_{i=1}^{N \times q}$. Thus, the objective is to classify the query set samples in each task, when we are provided with few support set samples from the target dataset.

**Prompt Learning:** Since *PromptMargin* is based on prompt learning, we briefly describe it here for completion. Prompt learning is an efficient and popular method of finetuning large-scale models like CLIP to downstream tasks, where a set of learnable vectors are appended to either the textual branch (Zhou et al., 2022b;a), or the visual branch (Jia et al., 2022), or both (Khattak et al., 2023). For our method, we use a multimodal prompt learning framework MaPLe (Khattak et al., 2023) as our baseline.

Let us denote the CLIP text encoder as $f_t$ and the image encoder as $f_v$. The input image $X \in \mathbb{R}^{C \times H \times W}$ is broken up into $M$ patches $\{e_1, e_2, ..., e_M\}$ and appended with the CLS token $e_{CLS}$ before passing it through the image encoder. Similarly, the text input, which is typically of the form "*a photo of a [CLASS]*", is embedded into the tokenized format $\{t_{SOS}, t_1, t_2, ..., c_k, t_{EOS}\}$ before passing it through the text encoder, where $t_1, t_2, ...$ denotes the token embeddings, $t_{SOS}$ and $t_{EOS}$ denotes the Start-of-Sentence and End-of-Sentence tokens respectively and $c_k$ denotes the kth classname. However, for multimodal prompt learning, we append both the text and visual inputs with learnable prompts. Specifically, let the $T$ learnable textual prompts be denoted as $\theta_t = \{\theta_{t_1}, \theta_{t_2}, ..., \theta_{t_T}\}$ and the $V$ learnable visual prompts be denoted as $\theta_v = \{\theta_{v_1}, \theta_{v_2}, ..., \theta_{v_V}\}$. In our setting, similar to Khattak et al. (2023), we project the textual prompts to visual prompts by a function $\mathcal{F}$, i.e., $\theta_v = \mathcal{F}(\theta_t)$. Then, the $\theta_t$ and $\theta_v$ are respectively appended to the text and vision inputs as follows: $X_T = \{t_{SOS}, \theta_t, t_1, t_2, ..., c_k, t_{EOS}\}$ and $X_V = \{e_{CLS}, \theta_v, e_1, e_2, ..., e_M\}$, before passing them through the text and image encoders. The final text and image embedding vectors can be written as $\tilde{X}_T = f_t(X_T)$ and $\tilde{X}_V = f_v(X_V)$. Apart from adding learnable prompt parameters to the input only (termed as *shallow prompting*), we also add such learnable prompts after every transformer block of the encoders (*deep prompting*) (Khattak et al., 2023). The final prediction is taken as the cosine similarity between the image and text embedding vectors. Finally, the multimodal prompts are jointly trained on a downstream classification task. Now we describe the proposed framework in detail.

## 5    Proposed PromptMargin Framework

The proposed PromptMargin works in a completely source-free setting, i.e. we directly finetune the CLIP model on the small number of samples provided in the support set of the target dataset. *We do not perform any meta-training on a separate source domain like mini-ImageNet as the existing state-of-the-art approaches (Song et al., 2024).* For this, we utilize prompt learning in both the textual and the visual branches, similar to Khattak et al. (2023) as described in the previous section. Given the support and query set from a randomly sampled episode, we train the prompts on the few samples available in the support set, and evaluate the model performance on the query set. In particular, suppose $(X, y) \in \mathcal{S}$, where $X \in \mathbb{R}^{k \times C \times H \times W}$ denotes the $k$ images in the support set, and $y \in \{c_1, c_2, ..., c_N\}$ denotes the $N$ classname texts. We append the textual and visual prompts to the classname texts and images respectively, and pass it through the CLIP encoders ($f_t$ and $f_v$). Let the final text and vision embedding vectors obtained be denoted as $\tilde{X}_T$ and $\tilde{X}_V$ (Sec. 4). Next, the prompts are learned through a cross-entropy loss objective, while the encoder parameters are kept frozen. The objective function can be written as follows:

$$\mathcal{L}_{CE} = \underset{\{\theta_t, \theta_v\}}{\operatorname{argmin}} \, \mathbb{E}_{(X,y) \sim \mathcal{S}} \, \mathcal{L}(sim(\tilde{X}_T, \tilde{X}_V), y) \tag{4}$$

where, $sim(.)$ denotes the cosine similarity. In this work, we aim to address the two important issues of this problem: (i) less data in the support set, (ii) unknown/specialized categories in the target domains. To address data scarcity, we use a **Selective Augmentation** strategy, where we only select the image augmentations, whose embeddings are close enough to the respective text embeddings in the joint representation space. In order to address the second challenge, we propose a **Multimodal Margin Regularizer**

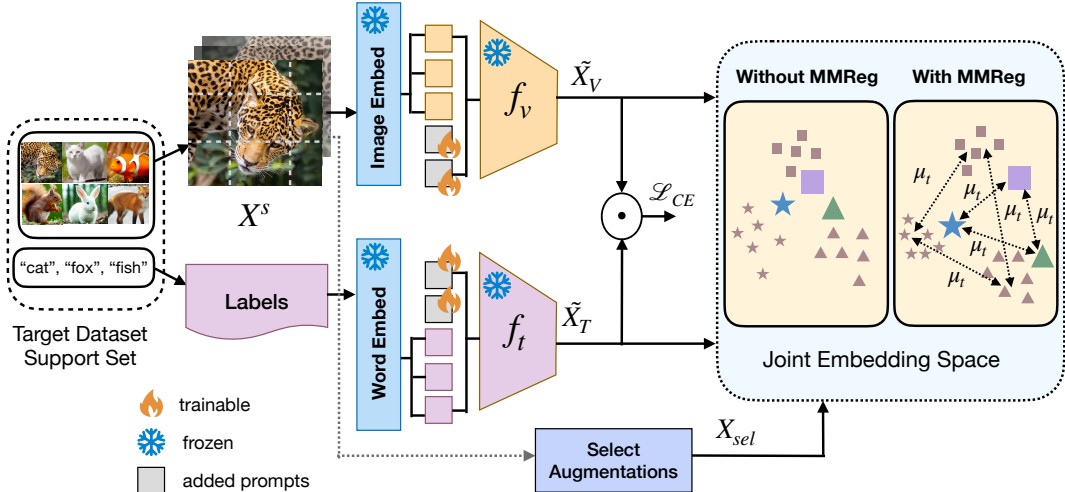

Figure 1: **An overview of our proposed _PromptMargin_ framework.** A randomly sampled episode from the target dataset is considered. The support set images along with their augmentations are passed through the CLIP image encoder, and their labels are passed though the CLIP text encoder. The selective augmentation strategy selects augmentations based on the embedding vectors. The Max-Margin Regularizer (MMReg) enforces the class-wise image prototypes and the text embeddings to uniformly separate out.

**(MMReg)** which uniformly separates the class-wise image and text prototypes in the joint feature space, thereby enforcing inter-class variability. The proposed framework is illustrated in Fig. 1. We now describe the two modules in detail.

## 5.1 Selective Augmentation

We use a selective augmentation strategy to increase the handful number of support set samples in the target dataset. For example, in the *5-way, 1-shot* setting, we have only 5 images (one image from each class) for the model to train on. Naturally, training the prompts on such less number of data samples can cause overfitting, hence, reducing the accuracy on the query set. To address this problem, we first take a combination of different augmentations of the support set images like HorizontalFlip, RandomRotation, ColorJitter, etc. However, instead of considering all the augmentations, we efficiently select only a few of the above augmented images, which works equally good, or better than taking all the augmentations, with a reduction in training time.

Let us consider the support set images as $X^s_{orig}$, and the respective augmented versions as $X^s_{aug}$. The total support samples can then be written as $X^s = \{X^s_{orig}; X^s_{aug}\}$. Consider the *5-way* setting. We have five classnames $(X_T)$, with which we append the learnable textual prompts and pass them through the frozen CLIP text encoder $f_t$, to obtain five text embeddings. Similarly, we append learnable visual prompts to $X^s$ and pass them through the CLIP image encoder $f_v$, and get the image embeddings. In the vision-language multimodal space, the $\mathcal{L}_{CE}$ tries to train the prompts such that the respective class text embeddings and the image prototypes (mean of the class-wise image embeddings) come closer. For the selective augmentation strategy, we choose a subset of $X^s$, whose cosine similarity with the corresponding class text embedding in the feature space is higher, i.e.,

$$X_{sel,r} = topr(sim(f_v([\theta_v; X^s]), f_t([\theta_t; X_T]))); \ s.t., X_{sel,r} \subset X^s. \tag{5}$$

Here, $topr(.)$ function takes the $r$ top values of its argument and $X_{sel,r}$ is the $r$ selected images from the set of all images $X^s$. Unlike scenarios where large number of training samples may be present and strong augmentations may be more beneficial, in this application with as few as one example per class, it is important that the augmentations are trained with class representative examples, which aids the model training.

Table 2: Effectiveness of the *selective augmentation* module. We generate different augmentations and select 15 examples based on the proposed strategy. We report accuracies and training time for the BSCDFSL benchmark after training on the same 20 sampled episodes.

| | EuroSAT | | ISIC | | Plant Disease | | Chest X-Ray | |
|---|---|---|---|---|---|---|---|---|
| Augmentations | Accuracy (%) | Time (mins.) | Accuracy (%) | Time (mins.) | Accuracy (%) | Time (mins.) | Accuracy (%) | Time (mins.) |
| 20 augs | 79.39 | 22.20 | 33.60 | 22.50 | 81.46 | 21.88 | 22.90 | 22.11 |
| 15 sel. augs | 78.26 | 18.23 | 34.40 | 18.23 | 78.00 | 18.58 | 23.00 | 18.57 |
| 30 augs | 79.20 | 30.03 | 33.20 | 30.48 | 80.27 | 29.97 | 20.60 | 30.06 |
| 15 sel. augs | 81.93 | 18.30 | 35.47 | 18.79 | 79.40 | 18.42 | 21.93 | 18.51 |
| 45 augs | 78.26 | 41.61 | 33.86 | 42.21 | 80.13 | 41.83 | 22.33 | 42.01 |
| 15 sel. augs | 78.47 | 18.35 | 35.33 | 18.95 | 77.66 | 18.55 | 22.53 | 18.56 |

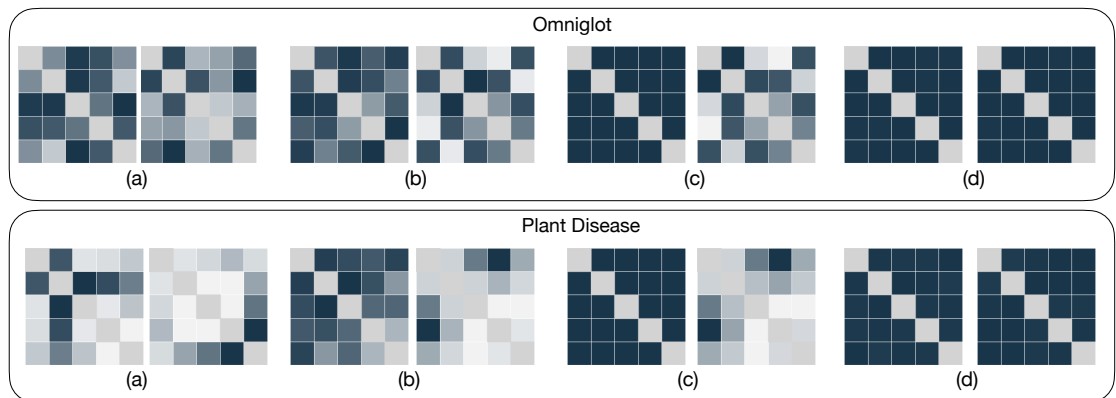

Figure 2: **Effectiveness of the *MMReg*.** All the heatmaps represent inter-class $L_2$ distances between embeddings for a representative episode in the 5-way setting. Darker hues represent higher values and vice versa. In all the images, the text features are followed by image features from left to right. (a) represents initial text and image feature distances, (b) represents the text and image embedding distances without MMReg, (c) represents the text and image distances with only text regularization, while (d) represents the text and image embeddings with MMReg. We observe in (b) that there is significant difference between the interclass distance of the image and text embeddings, implying that their embeddings are not that similar. Using only text regularization part separates the text but not the image features. In contrast, the maps are very similar for (d), which justifies the usefulness of the MMReg.

We perform a simple experiment to justify this selection strategy. Here, we initially generate different number of augmentations for each support image and then select 15 examples based on the proposed selection strategy. Table 2 shows the accuracies and training time for the four datasets in the BSCDFSL benchmark (Guo et al., 2020). We note that if the difference in the number of original and selected augmentations is low, the time and accuracy differences are not significantly impacted, and it may be feasible to consider all the original augmentations instead of a selection strategy. However, as the difference increases, it is worth considering this strategy due to time and accuracy considerations. Only for PlantDisease, we observe a slight decrease in performance, which upon analyzing the augmented images, we feel can be attributed to the quality of the initial augmentations. In conclusion, our proposed strategy reduces the training time by 40% while maintaining or even improving the accuracies consistently across the four datasets in majority of the cases.

## 5.2 Multimodal Margin Regularization

When deploying the model on a downstream task, the target dataset can contain examples of novel categories which are unseen and very different from what was encountered during pretraining. These data can be from specialized domains, e.g., EuroSAT (Helber et al., 2019) which contains satellite images, Chest X-Ray (Wang et al., 2017), containing medical X-Ray images, plant disease data (Mohanty et al., 2016), etc. Since

Table 3: Effect of prompt tuning with text regularization $\mathcal{R}(\tilde{X_T})$ vs MMReg.

| Method | Omniglot | Plantae | ISIC | Plant Disease |
|---|---|---|---|---|
| MaPLe + sel. augs | 85.49 | 64.69 | 33.54 | 82.66 |
| MaPLe + sel. augs + text reg | 85.76 | 64.65 | 33.44 | 82.54 |
| MaPLe + sel. augs + MMReg (PromptMargin) | **87.01** | **66.13** | **33.90** | **83.48** |

the CLIP model may not have seen similar data during training, it may not be able to generalize to these classes and bring their text and image embeddings close in the multimodal latent space using few labeled training examples. In addition, generalization of CLIP significantly depends on the presence of meaningful class names of the unseen categories. But often, such class names may not be provided, or even if they are available, they may not be semantically meaningful in the CLIP space. We have observed this in Section 3, which demonstrates a clear correspondence between zero-shot performance and the joint feature space alignments of the text and image embeddings. Thus, while finetuning with less amount of support set data, the model may not be able to generalize and discriminate between these classes. Now, we describe the proposed *Multimodal Margin Regularizer* (MMReg), which tries to simultaneously improve the inter-class discrimination and bring the inter-modal embeddings closer.

Our proposed MMReg is inspired from Hayat et al. (2019), where the regularizer addresses the training data imbalance problem in classification tasks by uniformly spreading out the classifier weights in the feature space. In our context, we aim to spread out the text embeddings in the feature space to avoid confusion between the distinct classes. The regularization term can be expressed as follows:

$$\mathcal{R}(\tilde{X_T}) = \frac{2}{N^2 - N} \sum_{i<j} (\|\tilde{X_{T_i}} - \tilde{X_{T_j}}\|_2^2 - \mu_t)^2, \ \forall j \in \{2, 3, ..., N\} \tag{6}$$

Here, $N$ denotes the number of classes in each episode, in a *N-way k-shot* setting. Each classname text, appended to learnable prompts is passed through the text encoder to obtain the classwise text embedding vectors $\tilde{X_T}$. The mean distance between these text embeddings is denoted by $\mu_t$ and can be written as:

$$\mu_t = \frac{2}{N^2 - N} \sum_{i<j} \|\tilde{X_{T_i}} - \tilde{X_{T_j}}\|_2^2, \ \forall j \in \{2, 3, ..., N\} \tag{7}$$

This regularizer trains the prompts in such a way that the text representations are uniformly separated by a distance of $\mu_t$. Since, the text prompts are coupled to the visual prompts through a function $\mathcal{F}(.)$, it is expected that this regularization term will also guide the visual representations to separate out. However, because of the presence of few training samples, we empirically observe that there is a lack of consistent separation between the image embedding prototypes compared to their textual counterparts. We consider four datasets to illustrate this in Table 3, where we observe that simply separating text features reduces the performance of prompt tuning (MaPLe). We can also visualize this from the $L_2$ distance heatmaps shown in Fig. 2 where the visual features are not affected by the text regularizer. Hence, we add another regularization term to the loss function, to separate out the image prototypes as follows:

$$\mathcal{R}(\tilde{X_V}) = \frac{2}{N^2 - N} \sum_{i<j} (\|\tilde{X_{V_i}} - \tilde{X_{V_j}}\|_2^2 - \mu_t)^2, \ \forall j \in \{2, 3, ..., N\} \tag{8}$$

where, $\tilde{X_V}$ are the prototypes (means) of the class image embeddings obtained from the visual encoder, i.e., $f_v([\theta_v; X_{sel}])$. $\mu_t$ is the same mean distance from Eqn. (7). This additional regularization term enforces the class-wise image prototypes to be equally separated by the same mean distance as the text representation embeddings. We observe in Table 3 that the proposed *MMReg* improves performance in all the cases. We also see this in Fig. 2 where the classname representations and the image prototypes are better separated in the feature space. Thus, the final loss function for training the prompts is given as:

$$\mathcal{L}_{total} = \mathcal{L}_{CE} + \mathcal{R}(\tilde{X_T}) + \mathcal{R}(\tilde{X_V}) \tag{9}$$

Minimizing this objective function ensures that the class text embeddings come close to the respective class image prototypes, and at the same time uniformly spreads out both of them in the joint vision-language representation space.

**Inference.** Once we train the joint VLM prompts on the support set image-text pairs of a sampled episode, we freeze the prompt parameters for the inference phase. Next, we feed the query set image-text pairs of that episode to our frozen model, and take the highest probability class as the final prediction.

## 6 Experimental Evaluation

Here, we describe the results of extensive experiments performed to evaluate the performance of the proposed approach. First, we describe the datasets used.

### 6.1 Dataset Description and Experimental Protocol

The proposed *PromptMargin* works in a source-free setting, i.e. the model is directly fine-tuned on the episodic support set of the target datasets. Thus, unlike many of the prior approaches, we do not require a source domain like ImageNet for pre-training our model. We conduct experiments on fifteen target benchmark datasets with varying distribution shifts, namely, EuroSAT (Helber et al., 2019), ISIC (Codella et al., 2019), Plant Disease (Mohanty et al., 2016), Chest X-Ray (Wang et al., 2017) (from BSCDFSL (Guo et al., 2020)), Omniglot (Lake et al., 2015), Traffic Signs (Houben et al., 2013), MSCOCO (Lin et al., 2014), Textures (Cimpoi et al., 2014), CUB (Wah et al., 2011), Quickdraw (Jongejan et al., 2016), Aircraft (Maji et al., 2013), VGG Flower (Nilsback & Zisserman, 2008), Fungi (Schroeder & Cui, 2018), mini-ImageNet (Vinyals et al., 2016) (from Metadataset (Triantafillou et al., 2019)), and the iNaturalist Plantae dataset (Van Horn et al., 2018). These datasets contain images ranging across different styles and categories, including medical images, satellite images, handwritten character images, etc.

As in the existing literature, we consider two settings for the experiments, namely, *5-way 1-shot* and *5-way 5-shot*, where one and five images from each of the five classes are randomly sampled in each episode for training. Following standard protocol (Guo et al., 2020; Liu et al., 2020; Liang et al., 2021), we also take 15 query images from the same set of classes, and evaluate the model on 600 episodes, and report the average accuracies and 95% confidence intervals.

**Implementation Details:** We use the CLIP ViT-B/16 backbone for all our experiments, similar to MaPLe (Khattak et al., 2023). For the learnable text prompts, we initialize the vectors with the standard text prompt *"A photo of a"*. The function $\mathcal{F}(.)$ is taken as a linear layer which projects the text prompts to visual prompts. The vision and text prompt lengths are set as 2. For *deep prompting*, we introduce learnable prompts before every transformer block upto a depth of 9. For the *1-shot* setting, we generate multiple augmentations, out of which we selectively choose 15 augmentations as discussed. Similarly, for the *5-shot* setting, we consider 3 selective augmentations, such that there are 15 examples per class. We jointly train the prompts on the support set images to minimize the final loss function, with SGD optimizer for 150 epochs with a learning rate of 0.01 and momentum of 0.9. Following Liang et al. (2021), we keep the above hyperparameters same across all datasets and settings. All the experiments are performed on a single NVIDIA RTX A5000 GPU. Now we report the results of experimental evaluation.

### 6.2 Comparison to the state-of-the-art methods

We compare our method with the most recent CLIP-based methods for all the fifteen datasets and report the results in Table 4. This includes full finetuned methods as well as prompt-tuning methods of CLIP. We also explicitly mention the backbones as well as different training procedures followed by the different approaches, which should be considered while comparing them.

A recent state-of-the-art approach, FDAlign (Song et al., 2024), was the first to investigate CLIP's generalization capability to the few-shot learning datasets with domain differences under similar settings. It proposes a CLIP finetuning technique, which is meta-trained on the miniImageNet dataset before adapting to the target datasets. Wise-FT (Wortsman et al., 2022) was originally proposed for the domain generaliza-

Table 4: The performances (accuracies) of CLIP-based methods Wise-FT, FDAlign, MaPLe and Ours (PromptMargin) on all the target datasets for both the 5-way 1-shot and 5-shot settings. Wise-FT and FD-Align has been trained on a source dataset while MaPLe and PromptMargin is directly trained on the few target samples. Highest values are marked in bold.

| Datasets | 5-way 1-shot | | | | 5-way 5-shot | | | |
|---|---|---|---|---|---|---|---|---|
| | WiSE-FT | FD-Align | MaPLe | PromptMargin | WiSE-FT | FD-Align | MaPLe | PromptMargin |
| EuroSAT | $63.99 \pm 0.39$ | $60.39 \pm 0.43$ | $75.46 \pm 0.19$ | $\mathbf{78.95} \pm 0.19$ | $80.96 \pm 0.19$ | $77.25 \pm 0.16$ | $89.55 \pm 0.10$ | $\mathbf{91.40} \pm 0.10$ |
| ISIC | $29.40 \pm 0.34$ | $28.84 \pm 0.44$ | $31.96 \pm 0.15$ | $\mathbf{33.90} \pm 0.15$ | $39.54 \pm 0.40$ | $38.91 \pm 0.44$ | $45.92 \pm 0.14$ | $\mathbf{46.88} \pm 0.16$ |
| Plant Disease | $75.66 \pm 0.33$ | $75.13 \pm 0.33$ | $79.38 \pm 0.22$ | $\mathbf{83.48} \pm 0.19$ | $91.78 \pm 0.31$ | $91.84 \pm 0.19$ | $93.32 \pm 0.11$ | $\mathbf{94.31} \pm 0.10$ |
| ChestX | $22.27 \pm 0.28$ | $\mathbf{22.31} \pm 0.17$ | $21.30 \pm 0.10$ | $21.51 \pm 0.10$ | $\mathbf{25.08} \pm 0.14$ | $24.95 \pm 0.15$ | $23.29 \pm 0.10$ | $23.92 \pm 0.09$ |
| iNaturalist Plantae | $-$ | $-$ | $55.34 \pm 0.29$ | $\mathbf{66.13} \pm 0.27$ | $-$ | $-$ | $83.07 \pm 0.26$ | $\mathbf{85.36} \pm 0.19$ |
| Omniglot | $83.56 \pm 0.28$ | $83.81 \pm 0.25$ | $77.82 \pm 0.29$ | $\mathbf{87.01} \pm 0.22$ | $95.26 \pm 0.09$ | $94.81 \pm 0.19$ | $96.23 \pm 0.10$ | $\mathbf{96.37} \pm 0.13$ |
| Traffic Signs | $60.84 \pm 0.29$ | $57.32 \pm 0.26$ | $56.45 \pm 0.24$ | $\mathbf{67.24} \pm 0.24$ | $78.11 \pm 0.24$ | $73.39 \pm 0.29$ | $85.21 \pm 0.19$ | $\mathbf{87.55} \pm 0.16$ |
| MSCOCO | $67.28 \pm 0.32$ | $\mathbf{69.16} \pm 0.28$ | $53.09 \pm 0.24$ | $56.12 \pm 0.24$ | $81.08 \pm 0.35$ | $\mathbf{81.37} \pm 0.24$ | $75.13 \pm 0.22$ | $78.68 \pm 0.23$ |
| Textures | $63.55 \pm 0.19$ | $66.05 \pm 0.12$ | $\mathbf{79.28} \pm 0.18$ | $78.99 \pm 0.20$ | $83.31 \pm 0.31$ | $83.60 \pm 0.34$ | $88.45 \pm 0.14$ | $\mathbf{88.71} \pm 0.15$ |
| CUB | $81.16 \pm 0.71$ | $82.38 \pm 0.69$ | $96.96 \pm 0.22$ | $\mathbf{96.97} \pm 0.22$ | $93.41 \pm 0.32$ | $93.87 \pm 0.24$ | $\mathbf{97.65} \pm 0.06$ | $97.12 \pm 0.06$ |
| Quickdraw | $62.54 \pm 0.59$ | $64.49 \pm 0.58$ | $72.54 \pm 0.22$ | $\mathbf{74.84} \pm 0.20$ | $82.78 \pm 0.37$ | $82.78 \pm 0.28$ | $85.08 \pm 0.14$ | $\mathbf{85.21} \pm 0.13$ |
| Aircraft | $62.64 \pm 0.62$ | $63.45 \pm 0.65$ | $\mathbf{79.76} \pm 0.27$ | $77.21 \pm 0.26$ | $77.66 \pm 0.59$ | $78.21 \pm 0.58$ | $\mathbf{87.56} \pm 0.21$ | $86.53 \pm 0.20$ |
| VGG Flower | $94.16 \pm 0.23$ | $93.50 \pm 0.24$ | $\mathbf{98.24} \pm 0.06$ | $97.65 \pm 0.07$ | $99.06 \pm 0.09$ | $98.95 \pm 0.09$ | $99.23 \pm 0.02$ | $\mathbf{99.27} \pm 0.02$ |
| Fungi | $53.10 \pm 0.27$ | $53.83 \pm 0.30$ | $58.55 \pm 0.27$ | $\mathbf{61.16} \pm 0.24$ | $73.28 \pm 0.10$ | $73.69 \pm 0.14$ | $79.69 \pm 0.20$ | $\mathbf{80.91} \pm 0.18$ |
| Mini-test | $93.55 \pm 0.17$ | $95.04 \pm 0.18$ | $\mathbf{99.17} \pm 0.03$ | $98.85 \pm 0.03$ | $98.44 \pm 0.06$ | $98.52 \pm 0.07$ | $\mathbf{99.39} \pm 0.02$ | $99.19 \pm 0.02$ |
| Average | $65.26$ | $65.40$ | $69.02$ | $\mathbf{72.00}$ | $78.55$ | $78.01$ | $81.92$ | $\mathbf{82.76}$ |

tion task, where the CLIP model was finetuned with parameter weight interpolations, which was adapted to the given setting. We compare with both of these methods, and include their performance accuracies as reported in Song et al. (2024). Additionally, for prompt learning methods we report MaPLe (Khattak et al., 2023), which is a vision-language prompt tuning method, which serves as another strong baseline for our method. We adapted this in our setting, where we finetune it directly on the few samples from the sampled episodes of the target datasets.

*Hence, we primarily explore whether CLIP can be deployed using prompt learning on very few samples on the target dataset without any meta-training on a large-scale dataset like ImageNet. Here, in addition to the challenge of robustly learning prompts with few samples, for many datasets, the class names are either not provided or are not quite meaningful for CLIP to generalize, thus making prompt-tuning even more challenging.*

For the *5-way 1-shot* setting, we observe that both full finetuning methods (FDAlign and Wise-FT) perform poorly compared to the parameter efficient finetuning methods. Our proposed method outperforms the baseline method MaPLe in eleven out of fifteen datasets, achieving an average accuracy of 72% compared to MaPLe's 69.02%. In some cases, where original classnames were not present or are semantically not meaningful, e.g., Plantae, Plant Disease, Traffic Signs, we achieve absolute accuracy gains of 10.79%, 4.10%, 10.79% respectively over MaPLe, highlighting the fact that our method gives significant improvement even when original classname texts are not present in the datasets. This is discussed in details further in the following section. However, our method exhibits slight decrements on certain datasets (like mini-ImageNet and Aircraft), which can be attributed to the dataset distributions being not so shifted in the CLIP space, as discussed later. For the *5-way 5-shot* setting, we observe a similar trend, where *PromptMargin* outperforms MaPLe in twelve out of fifteen datasets, achieving an average accuracy of 82.76%.

Our method aims to highlight that large-scale VLMs like CLIP can be efficiently transferred to out-of-distribution datasets with few-shot samples, without any access to source datasets, and can still provide a relatively close performance to meta-trained and finetuned methods.

## 6.3 Ablation Studies

Here, we analyze the effectiveness of the two proposed modules in PromptMargin, and summarize the results in Table 5. For analysis, we consider four representative datasets from across all the benchmarks and report accuracies for the *5-way 1-shot* setting using the same hyperparameters from Sec. 6.1 for 600 episodes. Since

Table 5: Ablation study: Both selective augmentation and MMReg are important.

| Method | Omniglot | Plantae | ISIC | Plant Disease |
|---|---|---|---|---|
| MaPLe (baseline) | 77.82 | 55.34 | 31.96 | 79.38 |
| MaPLe + MMReg | 78.46 | 58.04 | 32.49 | 79.59 |
| MaPLe + 15 sel augs | 85.49 | 64.69 | 33.54 | 82.66 |
| MaPLe + 15 sel augs + MMReg (PromptMargin) | **87.01** | **66.13** | **33.90** | **83.48** |

Table 6: Effect of the proposed regularization terms (MMReg) in separating out the interclass joint features in the CLIP representation space with only a single example per class.

| Dataset | EuroSAT | ISIC | Omniglot | Plant Disease | Quickdraw | Plantae | Traffic Signs | MSCOCO | mini-ImageNet |
|---|---|---|---|---|---|---|---|---|---|
| $m_T$ | 0.588 | 0.561 | 0.679 | 0.770 | 0.716 | 0.100 | 0.100 | 0.100 | 0.860 |
| $m_V$ | 0.627 | 0.582 | 0.420 | 0.540 | 0.520 | 0.727 | 0.583 | 1.010 | 1.010 |
| MaPLe (%) | 75.46 | 31.96 | 77.82 | 79.38 | 72.54 | 55.34 | 56.45 | **53.09** | **99.17** |
| MaPLe + MMReg (%) | **75.81** | **32.49** | **78.46** | **79.59** | **73.02** | **58.04** | **59.24** | 51.90 | 98.99 |

our framework is built upon MaPLe (Khattak et al., 2023), we report its accuracies as the baseline method. As illustrated in the table, both the proposed modules improve the baseline method significantly.

We had proposed the MMReg module based on the observations of the feature alignments in the joint CLIP vision-language space. Now, we see in Table 6 how this simple regularization term is effective in guiding the vision language features to separate out even for a single example per class. As noted in Section 3, when the inter-class text and image embedding distances ($m_T$ and $m_V$) were low, prompt-learning (MaPLe) had improved the performance, but additionally incorporating our regularizer further improves the class discriminations and results in better accuracy. Notably, we observe that when the image feature separation ($m_V$) is low, and placeholder classnames have been used, the improvement with our MMReg module is significant (+2.7 and +2.79 for Plantae and Traffic Signs respectively). Although in MSCOCO, pseudo-classnames have been used, the image features are extremely well separated, resulting in the MMReg module slightly decreasing the performance. However, utilizing only the text regularization term $\mathcal{R}(\tilde{X}_T)$ improves the prompt-learning accuracy by +1.28, hence conforming with our proposed notion of separating out closely situated embeddings in the representation space. For mini-ImageNet, since both modalities are well separated, prompt-learning, even with our regularizer, on the few samples does not improve results, and can adversely affect the latent space alignment. We also illustrate some qualitative results in Fig. 3 for visualization of some of the different datasets.

**Scope for future work:** Although the two modules of our proposed framework demonstrate good performance for most of the datasets, in few cases, it failed to outperform MaPLe. As for the first module, a lack of carefully chosen augmentation strategies may hurt the generalization more than it improves. As an example, in Fig. 4, we illustrate some poor augmentations generated for the Aircraft dataset, where our method fails to improve over MaPLe. Similarly, in some cases, where the features in the latent space are already well separated, further finetuning with MMReg may adversely affect the CLIP space, hence reducing the performance. Nevertheless, this work may serve as a strong baseline for robust prompt learning techniques of foundation models like CLIP for such challenging and real-world settings.

# 7    Conclusion

Large-scale vision-language models like CLIP are emerging as a popular choice due to their powerful zero-shot generalization capabilities. Prompt learning is an efficient technique to transfer CLIP-like models to downstream datasets with few samples. However, to the best of our knowledge, there has been no work where prompt learning has been utilized for classification tasks where the datasets simultaneously contain few samples as well as a shift in distribution from natural images. In this work, we explore the possibility of learning only a few prompt parameters on the target datasets, in a completely source-free manner. Extensive

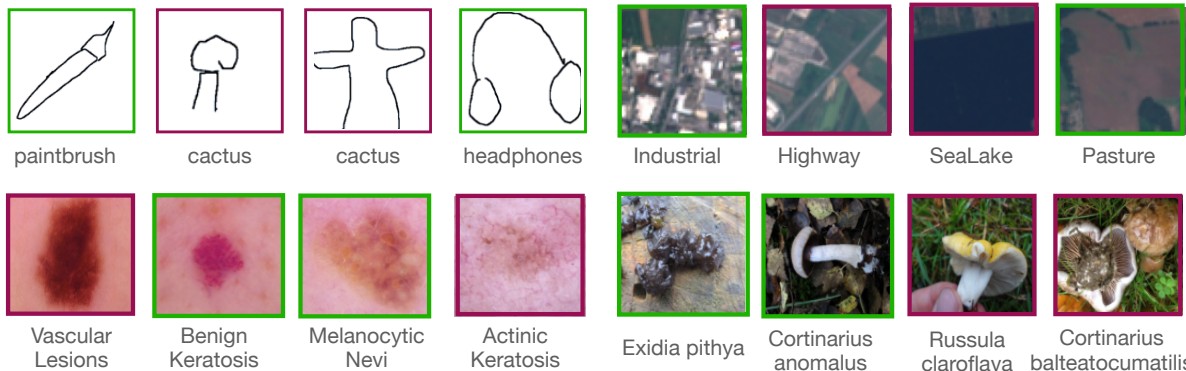

Figure 3: **Some qualitative results across different datasets.** From top left, samples are shown from Quickdraw (Jongejan et al., 2016), EuroSAT (Helber et al., 2019), ISIC (Codella et al., 2019) and Fungi (Schroeder & Cui, 2018) datasets. Green and red denote correct and incorrect predictions respectively.

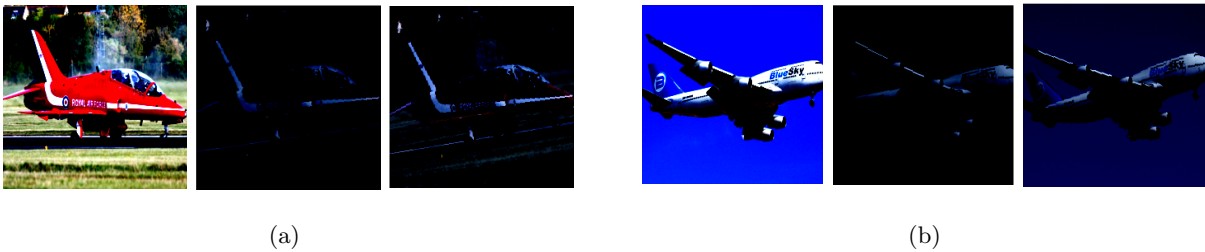

Figure 4: Some instances of poor augmentations generated for two support set images, leading to reduced generalization. (a) and (b) represents images from two classes of the Aircraft dataset, namely "Hawk T1" and "Boeing 747-400".

experiments on standard benchmark datasets highlight the efficacy of our proposed approach over state-of-the-art methods.

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

# A   Choice of the number of Selective Augmentations

In general, generating more augmentations can improve performance by mitigating overfitting in few-shot learning, but these can also include bad augmentations. The proposed Selective Augmentation module aims to efficiently select a subset of the generated augmentations, removing worse augmentations in the feature space, and hence maintaining a competitive performance while reducing the training time significantly as discussed in Section 5.1. Here, we report a sensitivity analysis on the number of selective augmentations in Table 7 for two datasets, EuroSAT and ISIC. As expected, for very few augmentations (say, 5), due to less number of examples, the results are in general lower. After around 10 augmentations, the performance stabilizes and does not vary significantly with the number of augmentations. We have chosen 15 augmentations in all our experiments to achieve good performance in reasonable training time, since for few-shot setting, it is not possible to tune hyperparameters for individual datasets. Dynamically choosing the number of selective augmentations in an episodic manner can be an interesting direction for future work.

# B   Analysis of Selective Augmentations

We first randomly generate a large number of augmentations using standard techniques like RandomHorizontalFlip, RandomResizedCrop, ColorJitter, etc. The initially generated augmented images are a random mixture of the various augmentation strategies and produces a variety of augmentations which differ across episodes as well as datasets. The selective augmentation module then selects 15 augmentations from these samples based on their cosine similarity with the corresponding class text embeddings in the feature space. To understand which augmentations are being chosen implicitly, we perform an experiment in which instead of taking random augmentations, we fix the number of augmentations of each type. E.g., we generate 6 images by RandomCropping, followed by 6 images by ColorJitter and so on, and then analyze which are being discarded by this proposed module. We observe that in general, the augmentations generated by RandomCropping are discarded, where the region of interest is being removed, followed by ColorJitter, which sometimes blurs or blackens the images. Analysis on EuroSAT and ISIC datasets is illustrated in Fig. 5.

Table 7: Sensitivity analysis on selective augmentations for the same 20 sampled episodes.

| Augmentations | EuroSAT Accuracy (%) | ISIC Accuracy (%) |
|---|---|---|
| 20 augs | 79.39 | 33.60 |
| 5 sel. augs | 75.53 | 28.13 |
| 10 sel. augs | 81.47 | 34.47 |
| 15 sel. augs | 78.26 | 34.40 |
| 30 augs | 79.20 | 33.20 |
| 5 sel. augs | 76.53 | 27.07 |
| 10 sel. augs | 77.47 | 36.27 |
| 15 sel. augs | 81.93 | 35.47 |
| 20 sel. augs | 79.53 | 33.86 |
| 45 augs | 78.26 | 33.86 |
| 5 sel. augs | 79.47 | 26.73 |
| 10 sel. augs | 79.53 | 34.40 |
| 15 sel. augs | 78.47 | 35.33 |
| 20 sel. augs | 79.40 | 35.40 |
| 60 augs | 78.73 | 33.26 |
| 5 sel. augs | 75.13 | 25.86 |
| 10 sel. augs | 77.67 | 29.80 |
| 15 sel. augs | 79.73 | 33.66 |
| 20 sel. augs | 80.33 | 28.73 |

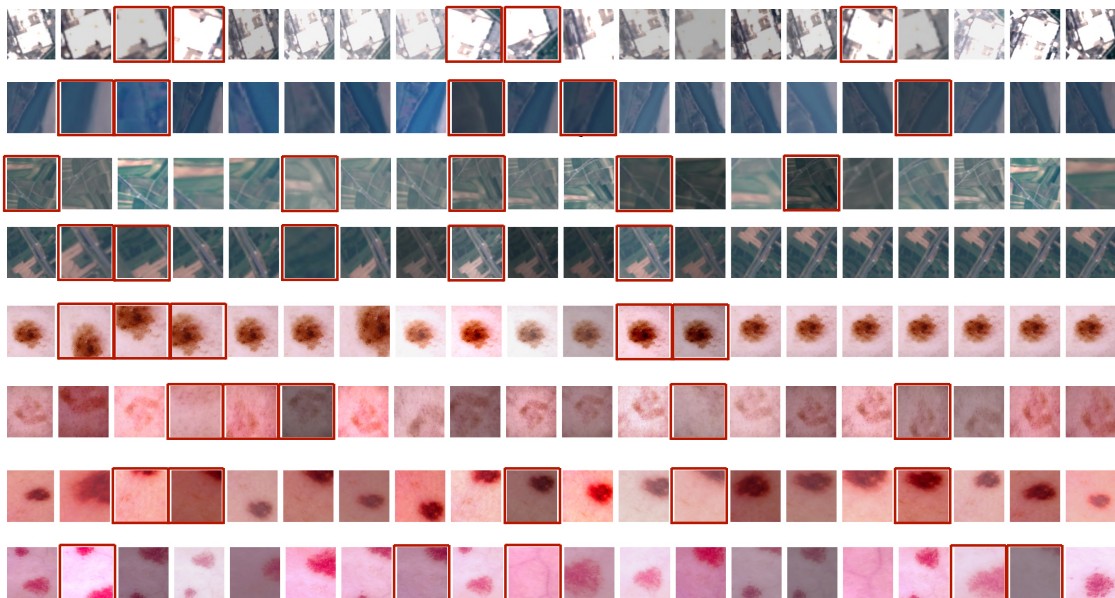

Figure 5: Visualizations of the augmentations being selected by the proposed Selective Augmentation module for EuroSAT and ISIC with 20 initial augmentations. Each row corresponds to a particular class. The red borders denote the ones which are discarded. We observe that augmentations where the region of interest are removed or darkened, have fairly more chance of getting discarded. However, in some cases, more augmentations may be needed to be removed than only five.

Table 8: Effect of strong vs weak augmentations.

|  | **EuroSAT** | **ISIC** |
| Dataset | Accuracy (%) | Accuracy (%) |
| --- | --- | --- |
| 20 standard augs | 79.39 | 33.60 |
| 20 strong augs | 77.93 | 34.93 |
| 30 standard augs | 79.20 | 33.20 |
| 30 strong augs | 77.20 | 32.33 |
| 45 standard augs | 78.26 | 33.86 |
| 45 strong augs | 77.60 | 30.67 |

## C  Weak vs strong augmentations

Strong augmentations are intense transformations of the images like AutoAugment and CutOut, whereas weak augmentations are simple transformations like Flip and Shift. In our work, we have used standard augmentations like RandomCrop and HorizontalFlip, which can be attributed to weak augmentations, from which we have then selected a subset based on the feature space similarities. To understand the effect of strong augmentations instead of standard augmentations (as originally used), we consider two benchmark datasets, EuroSAT and ISIC and apply RandomErasing (which removes a tile from the image similar to CutOut), and AutoAugment. The results in Table 8 suggest that, in general, such aggressive augmentation strategies do not help in classification as it distorts and removes crucial information from the images, thereby hurting the generalization. In very few instances, such strong augmentations can crop the background and focus on the region of interest in datasets like ISIC. Some visualizations are shown in Figs. 7 and 8.

Table 9: Experiments on OpenCLIP B/16.

| Dataset | Omniglot | Quickdraw | EuroSAT | Plant Disease | ChestX |
|---|---|---|---|---|---|
| Zero-shot (%) | 27.79 | 66.49 | 48.93 | 24.65 | 20.08 |
| MaPLe (%) | 80.61 | 72.21 | 71.60 | 77.50 | 21.09 |
| PromptMargin (%) | **89.44** | **77.08** | **73.28** | **82.26** | **22.75** |

Table 10: Effect of MMReg when classnames are not available.

| Dataset | Plantae | ISIC | ChestX |
|---|---|---|---|
| MaPLe (%) (with classnames) | - | 31.96 | 21.30 |
| MaPLe (%) (with pseudo classnames) | 55.34 | 28.32 | 20.19 |
| MaPLe + MMReg (%) | **58.04** | **30.24** | **20.89** |

## D  Experiments on an alternate VLM

We have reported results for CLIP ViT-B/16 in the main text for fair comparison to other prompt tuning methods. Here, we consider an alternate VLM, OpenCLIP B/16 (Cherti et al., 2023), which has been pretrained on a different dataset. The results on some benchmark datasets (with the same hyperparameters) are reported in Table 9. We observe that although MaPLe improves over the zero-shot performance, PromptMargin outperforms MaPLe in all the datasets, further justifying the effectiveness of the proposed framework.

## E  t-SNE Visualizations

The t-SNE visualizations of the image embeddings after training with our proposed MMReg module is shown in Fig. 6 for two benchmark datasets, Omniglot and Plant Disease. We observe that training with the MMReg regularizer uniformly separates and more compactly clusters the few-shot image examples in the feature space.

## F  Effect of MMReg in the absence of classnames

Since in many specialized datasets (e.g., medical datasets like ISIC and Chest X-Ray), meaningful classnames have been used, the model may still rely on them for proper classification in the CLIP space. However, often classnames may not be available for such datasets in practical settings. To emulate such a scenario where classnames may not be available, we replace the classnames of these specialized datasets with pseudo classnames (like C1, C2, C3, etc.), and see the effect of the MMReg module on them (Table 10). We observe that while performance of MaPLe (prompt tuning) drops because the class semantics are now missing, our MMReg module helps them separate out in the multimodal representation space, thereby improving the performance in both the datasets, which conforms with our motivation for this proposed module.

## G  Sensitivity analysis

In Table 11, we vary the weighting coefficients ($\alpha$ and $\beta$) of the two terms of the MMReg module ($\mathcal{R}(\tilde{X}_T)$ and $\mathcal{R}(\tilde{X}_V)$) respectively for two benchmark datasets, EuroSAT and ISIC. In the main text, we have used $\alpha = 1$ and $\beta = 1$ for all experiments, since hyperparameter tuning is not possible for few-shot learning. But we observe that the performance is quite stable for a wide range of parameters.

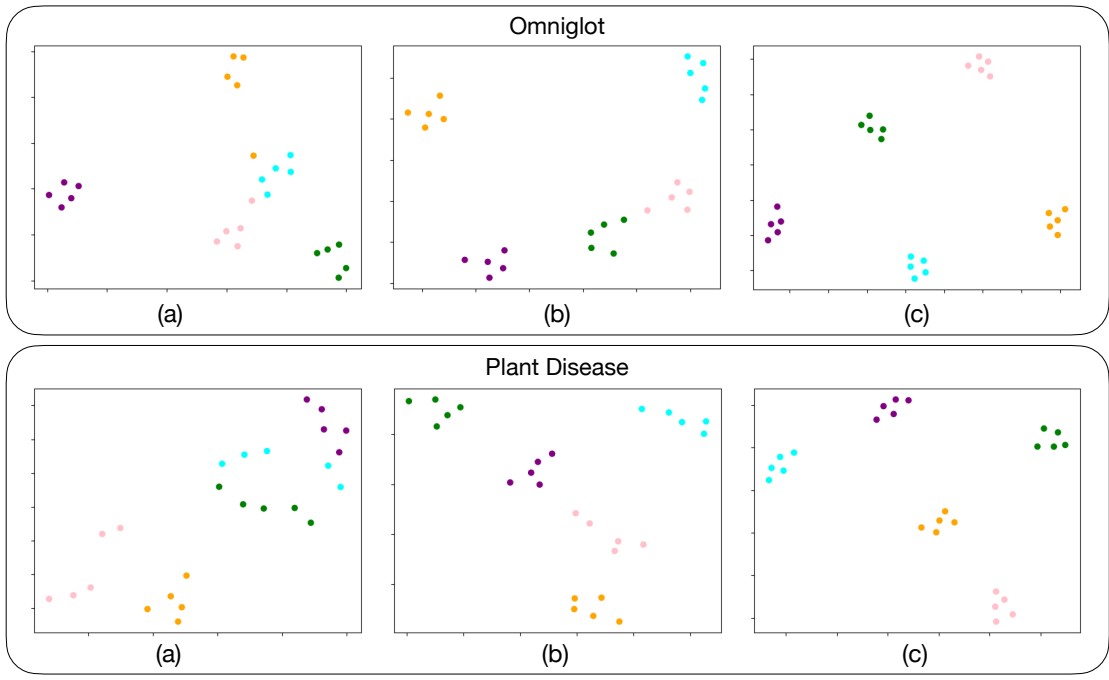

Figure 6: Visualizations of the image embeddings using t-SNE for the 5-way setting. (a) denotes the image features initially, (b) trained with MaPLe and (c) with the proposed MMReg. We see that MMReg more compactly clusters and uniformly separates the embeddings in the feature space compared to normal training.

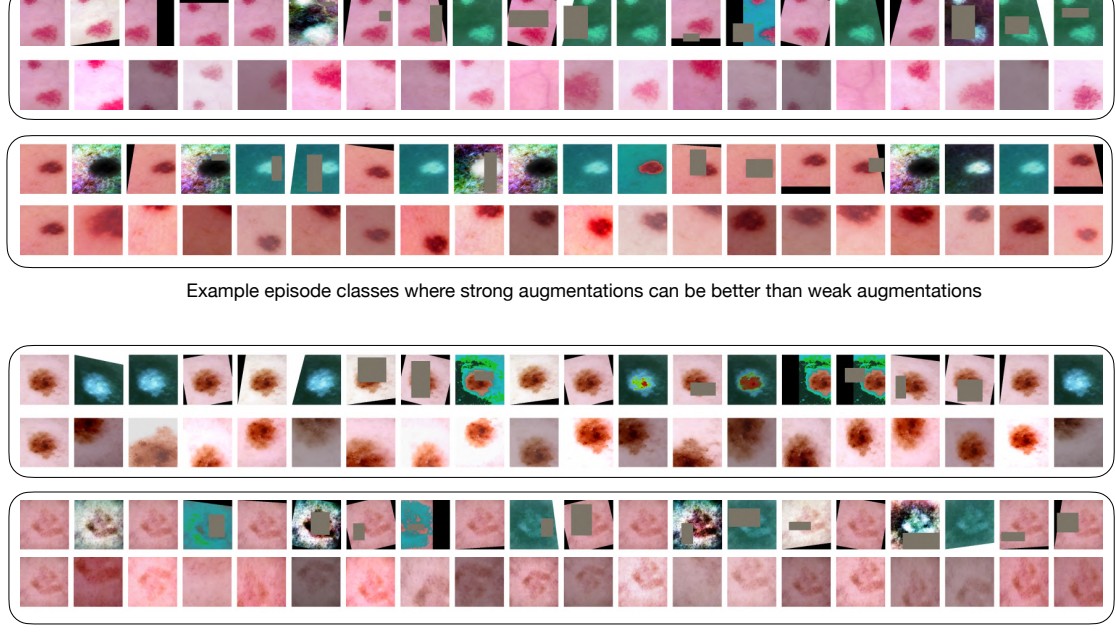

Figure 7: Visualizations of strong augmentations and weak augmentations for the ISIC dataset. Each block corresponds to a particular class. The first and the second rows in each block corresponds to strong and weak augmentations respectively. In some instances, the strong augmentatioons can be better than weak augmentations and vice versa.

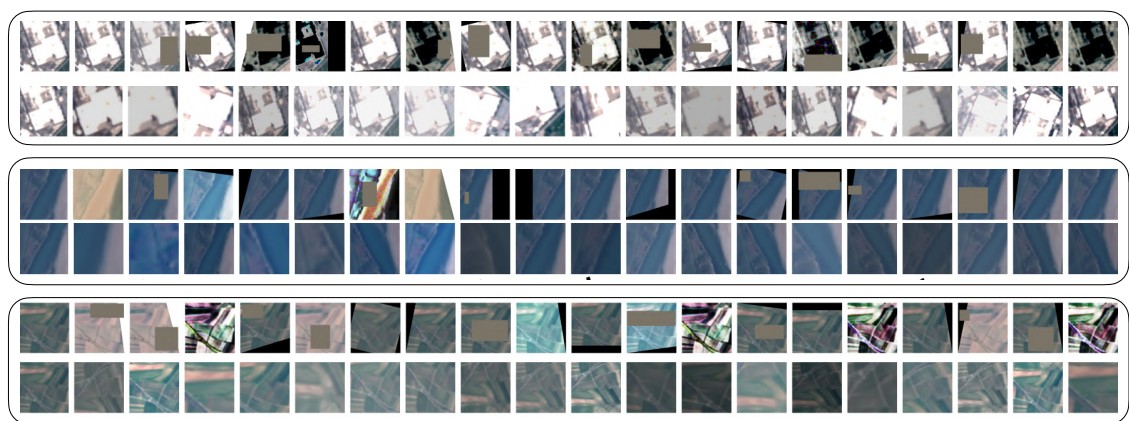

Weak augmentations are better than strong augmentations

Figure 8: Visualizations of strong augmentations and weak augmentations for the EuroSAT dataset. Here, strong augmentations are always worse than weak augmentations since the region of interest is not confined to a specific region.

| Dataset | Weighting Coefficients | | Accuracy |
|---------|---------|---------|---------|
| | $\alpha$ | $\beta$ | |
| | 1 | 1 | 76.33 $\pm$ 0.095 |
| EuroSAT | 1 | 2 | 77.61$\pm$ 0.019 |
| | 2 | 1 | 77.49$\pm$ 0.018 |
| | 1 | 1 | 32.93$\pm$0.064 |
| ISIC | 1 | 2 | 32.77$\pm$0.062 |
| | 2 | 1 | 32.88$\pm$ 0.063 |

Table 11: Hyperparameter sensitivity analysis for $\alpha$ (for $\mathcal{R}(\tilde{X}_T)$) and $\beta$ (for $\mathcal{R}(\tilde{X}_V)$).

