# OpenReview forum: "Prompt Tuning Vision Language Models with Margin Regularizer for Few-Shot Learning under Distribution Shifts"
_TMLR — Accepted by TMLR_

### Review · Reviewer_8apX · 2024-10-23

**Summary Of Contributions:**

The paper presents a multimodal prompt tuning method for few-shot learning. First, the authors show that the mean distance between the image and the text embeddings from the CLIP model is a simple but unique approach to determining whether fine-tuning CLIP improves or hurts model performance. Then, they introduce the PromptMargin framework, a multi-modal prompt tuning method that includes selective data augmentation and multi-modal margin regularization.
The results for few-shot adaptation (1-shot and 5-shot) show improved performance, on average, compared to strong baselines on 15 datasets.

**Audience:**

Yes

**Claims And Evidence:**

Yes

**Requested Changes:**

Please see the weaknesses.

**Strengths And Weaknesses:**

### Strengths
- The paper is well-written. The motivation is very clear from the abstract. The method is clear. The experiment show that the PromptMargin framework indeed improves over the baseline methods.
- In section 3, the authors analyzed the mean distances between inter-class prototypes and found that when the target dataset has higher differences show the lowest performance with vanilla CLIP. On the other hand, if the differences are low, then the model performance after fine-tuning vanilla CLIP can also drop. This is an interesting insight. This adds to the strength of the authors’ claims.
- The ablation studies show that selective augmentation boosts the performance the most. This is an interesting insight.

### Weaknesses
- While selective augmentation is a neat trick to boost performance, the paper does not provide a clear understanding of why it works.There are several unanswered questions: (a) why did the authors choose 15 augmentations as opposed to 20?, (b) are the 15 augmentations same across the datasets?, and (c) how is this related to the weak augmentations commonly used in self training [a]. A more detailed analysis and experiments are required to understand the effect of selective data augmentation.

- The work is limited only to CLIP. It will be very interesting to readers if the same conclusions from the experiments hold true for other vision-language models.

### Minor
- In Section 3, the authors claim they estimate the extent of the distribution shift. While the results empirically demonstrate their claim, I would like to see the Pearson correlation between CLIP performance and the distribution and semantic differences metrics.

- Tables 1 and 2 are a bit difficult to understand at first glance because the first three rows are different from the last two rows, and the units are not comparable. The authors could separate the tables.

- Table 4 bolds results from row 2 and row 4. However, only row 4 has the best accuracy, and row 3 has the second-best. Please correct the table accordingly.

[a] FixMatch: Simplifying Semi-Supervised Learning with Consistency and Confidence. NeurIPS 2020

---

> ### Author Response · Authors · 2024-11-06
> **Response to Reviewer 8apX (1/2)**
>
> We thank the reviewer for appreciating the clarity in the motivation and method of our work. We now address the concerns.
> >Why did the authors choose 15 augmentations as opposed to 20?
>
> In general, generating more augmentations can improve performance by mitigating overfitting in few-shot learning, but these can also include bad augmentations. The proposed Selective Augmentation module aims to efficiently select a subset of the generated augmentations, removing worse augmentations in the feature space, and hence maintaining a competitive performance while reducing the training time significantly. We report the performance on two representative datasets, EuroSAT and ISIC, by varying the number of selective augmentations instead of fixing it to 15 (Table A here). As expected, for very few augmentations (say, 5), due to less number of examples, the results are in general lower. After around 10 augmentations, the performance stabilizes and does not vary significantly with the number of augmentations. We have chosen 15 augmentations in all our experiments to achieve good performance in reasonable training time, since for few-shot setting, it is not possible to tune hyperparameters for individual datasets. However, dynamically choosing the number of selective augmentations in an episodic manner can be an interesting direction for future work.
>
> |               | EuroSAT | ISIC |
> |---------------|--------------------------------------|-----------------------------------|
> | Augmentations | Accuracy (\%)                        | Accuracy (\%)                     |
> | 20 augs       | 79.39                                | 33.60                             |
> | 5 sel. augs   | 75.53                                | 28.13                             |
> | 10 sel. augs  | 81.47                                | 34.47                             |
> | 15 sel. augs  | 78.26                                | 34.40                             |
> | | | |
> | 30 augs       | 79.20                                | 33.20                             |
> | 5 sel. augs   | 76.53                                | 27.07                             |
> | 10 sel. augs  | 77.47                                | 36.27                             |
> | 15 sel. augs  | 81.93                                | 35.47                             |
> | 20 sel. augs  | 79.53                                | 33.86                             |
> | | | |
> | 45 augs       | 78.26                                | 33.86                             |
> | 5 sel. augs   | 79.47                                | 26.73                             |
> | 10 sel. augs  | 79.53                                | 34.40                             |
> | 15 sel. augs  | 78.47                                | 35.33                             |
> | 20 sel. augs  | 79.40                                | 35.40                             |
> | | | |
> | 60 augs       | 78.73                                | 33.26                             |
> | 5 sel. augs   | 75.13                                | 25.86                             |
> | 10 sel. augs  | 77.67                                | 29.80                             |
> | 15 sel. augs  | 79.73                                | 33.66                             |
> | 20 sel. augs  | 80.33                                | 28.73                             |
>
> TABLE A: Sensitivity analysis on selective augmentations for the same 20 sampled episodes.
>
> >Are the 15 augmentations same across the datasets?
>
> We first randomly generate a large number of augmentations using standard techniques like RandomHorizontalFlip, RandomResizedCrop, ColorJitter, etc. The initially generated augmented images are a random mixture of the various augmentation strategies and produces a variety of random augmentations which differ across episodes as well as datasets. Our approach then selects 15 augmentations from these samples based on their cosine similarity with the corresponding class text embeddings in the feature space. To understand which augmentations are being chosen implicitly, we perform an experiment in which instead of taking random augmentations, we fix the number of augmentations of each type. For instance, we generate 6 images by RandomCropping, followed by 6 images by ColorJitter and so on, and then analyze which are being discarded by this proposed module. We observe that in general, the augmentations generated by RandomCropping are discarded, where the region of interest is being removed, followed by ColorJitter, which sometimes blurs or blackens the images. This analysis on EuroSAT and ISIC datasets is included in the Appendix (Figure 7) of our paper.

---

> > ### Author Response · Authors · 2024-11-06
> > **Response to Reviewer 8apX (2/2)**
> >
> > >How is this related to the weak augmentations commonly used in self training?
> >
> > In [1], the strong augmentations are referred to as intense transformations to the images like AutoAugment and CutOut, whereas weak augmentations are simple transformations like Flip and Shift. In our work, we have used standard augmentations like RandomCrop and HorizontalFlip, which can be attributed to weak augmentations, from which we have then selected a subset based on the feature space similarities. Our contribution lies in the selection strategy rather than the type of augmentations used. To understand the effect of strong augmentations instead of standard augmentations (as originally used), we consider two benchmark datasets, EuroSAT and ISIC and apply RandomErasing (which removes a tile from the image similar to CutOut), and AutoAugment. We observe from the results (Table B here), that in general, such aggressive augmentation strategies doesn't help in classification as it distorts and removes crucial information from the images, thereby hurting the generalization. In very few instances, such strong augmentations can crop the background and focus on the region of interest in datasets like ISIC. For visualization, we have added the qualitative results in the Appendix (Figure 8 and 9).
> >
> > |                  | EuroSAT | ISIC |
> > |------------------|--------------------------------------|-----------------------------------|
> > |   Augmentation type    | Accuracy (\%)                        | Accuracy (\%)                     |
> > | | | |
> > | 20 standard augs | 79.39                                | 33.60                             |
> > | 20 strong augs   | 77.93                                | 34.93                             |
> > | | | |
> > | 30 standard augs | 79.20                                | 33.20                             |
> > | 30 strong augs   | 77.20                                | 32.33                             |
> > | | | |
> > | 45 standard augs | 78.26                                | 33.86                             |
> > | 45 strong augs   | 77.60                                | 30.67                             |
> >
> > TABLE B: Effect of strong vs weak augmentations.
> > >The work is limited only to CLIP. It will be very interesting to readers if the same conclusions from the experiments hold true for other vision-language models.
> >
> > Thank you for the suggestion. Prior literature on Prompt tuning mainly employs CLIP ViT-B/16 as the preferred backbone. Since our method is built on a prompt tuning framework, we have utilized the same architecture for all the experiments for a fair comparison. As suggested, we consider an alternate VLM, OpenCLIP B/16 [2], which has been pretrained on a different dataset. The results on some benchmark datasets (with the same hyperparameters) are reported here (Table C). We observe that although Maple improves over the zero-shot performance, PromptMargin outperforms MaPLe in all the datasets, further justifying the effectiveness of the proposed framework.
> >
> > | Dataset           | Omniglot | Quickdraw | EuroSAT | Plant Disease | ChestX |
> > |-------------------|----------|-----------|---------|---------------|--------|
> > | Zero-shot (\%)    | 27.79    | 66.49     | 48.93   | 24.65         | 20.08  |
> > | MaPLe (\%)        | 80.61    | 72.21     | 71.60   | 77.50         | 21.09  |
> > | PromptMargin (\%) | **89.44**    | **77.08**     | **73.28**   | **82.26**   | **22.75**  |
> >
> > TABLE C: Experiments on OpenCLIP B/16.
> > >In Section 3, the authors claim they estimate the extent of the distribution shift. While the results empirically demonstrate their claim, I would like to see the Pearson correlation between CLIP performance and the distribution and semantic differences metrics.
> >
> > As suggested, we compute the Pearson correlation between the CLIP performance and the distribution and semantic differences metric $diff(m_T, m_V)$, which comes out to be $-0.713$, and shows a strong negative correlation. This conforms with our analysis which suggests that an increase in the proposed domain difference metric results in reduced zero-shot performance of CLIP.
> > >Tables 1 and 2 are a bit difficult to understand at first glance because the first three rows are different from the last two rows, and the units are not comparable. The authors could separate the tables. Table 4 bolds results from row 2 and row 4. However, only row 4 has the best accuracy, and row 3 has the second-best. Please correct the table accordingly.
> >
> > We thank the reviewer for the suggested changes. We will correct them in the revised manuscript.
> >
> > [1] Sohn, Kihyuk, et al., "FixMatch: Simplifying Semi-Supervised Learning with Consistency and Confidence", NeurIPS (2020).
> >
> > [2] Cherti, Mehdi, et. al., "Reproducible scaling laws for contrastive language-image learning", CVPR (2023).

---

### Review · Reviewer_DSBP · 2024-10-26

**Summary Of Contributions:**

1. Empirical Analysis: Explored CLIP's zero-shot performance on various target datasets by analyzing image and text feature distances in multimodal space.
2. PromptMargin Framework: Introduced a novel prompt learning framework that adapts CLIP for few-shot tasks with distribution shifts, without requiring training on a source dataset.
3. Multimodal Margin Regularizer (MMReg): Developed MMReg to improve prompt learning by ensuring uniform separation of image and text category embeddings.
4. Performance Evaluation: Demonstrated superior performance across 15 benchmark datasets in 1-shot and 5-shot settings, compared to zero-shot, meta-training, and baseline methods.

**Audience:**

No

**Claims And Evidence:**

Yes

**Requested Changes:**

Please refer to weakness.

**Strengths And Weaknesses:**

Strengths: please refer to the contributions

Weaknesses:

1. The motivation of the proposed methods is not strong enough. While I agree that improving the inter-class variability and selecting only the most relevant augmentations based on their similarity is intuitively helpful, it is widely discussed in traditional domain adaptation literatures and this paper does a simple adaptation in the context of CLIP. I fail to find new insights in the existing submission. Are there any evidences that the mentioned issues (and the corresponding solutions) are especially important for CLIP?

2. While the MMReg aims to improve generalization by separating class embeddings, the model’s ability to handle highly specialized or unfamiliar domains (e.g., medical images) still relies on the presence of meaningful class names, which may not always be available.

---

> ### Author Response · Authors · 2024-11-06
> **Response to Reviewer DSBP (1/2)**
>
> We thank the reviewer for the thoughtful comments. We address the concerns as follows:
> >The motivation of the proposed methods is not strong enough. While I agree that improving the inter-class variability and selecting only the most relevant augmentations based on their similarity is intuitively helpful, it is widely discussed in traditional domain adaptation literatures and this paper does a simple adaptation in the context of CLIP. I fail to find new insights in the existing submission. Are there any evidences that the mentioned issues (and the corresponding solutions) are especially important for CLIP?
>
> Although we agree that the modules in our proposed framework have been inspired from ideas from prior literature in other contexts, we summarize the importance of the issues and our contributions in the context of CLIP as follows:
> - **Motivation of the addressed problem:** CLIP performs very poorly on OOD or specialized datasets like satellite and medical images, as noted in the original paper [1], which we also observe in our work (EuroSAT (47.7\%), ISIC (22.40\%), Plantae (26.54\%) in Table 1). Here, we consider a realistic setting where a few training samples are present from such datasets, on which we can adapt the pretrained CLIP model. Recently, CLIP finetuning approaches like Prompt Tuning have been explored for few-shot learning, however they do not consider extreme dataset distribution shifts. Other works train CLIP on a large source dataset (like ImageNet) before adapting them to such settings. We try to address this gap, by trying to adapt CLIP to such challenging scenarios without further training on a source dataset.
> - **Contribution I:** We analyze the CLIP embedding space with the few available samples and design a distribution and semantic difference metric, which gives an estimate of how the model would work on different datasets having domain shifts. To the best of our knowledge, this has not been explored in prior literature, and we feel that this provides an interesting insight into analyzing such pretrained VLMs, where we do not have knowledge about the pretraining dataset.
> - **Contribution II:** Generating augmentations help few-shot training by mitigating the overfitting problem. However, bad augmentations can overshadow the effect of good samples, hence we propose the selective augmentation strategy. For single modality,  image features and the classifiers are learnt together using the training data. However, in the context of multimodal models like CLIP, the text features (classifiers) and the image features are learnt separately, so that they come close in the multimodal feature space during finetuning. For this, we consider only the most similar augmentations to the text classifiers in the *feature space, hence reducing the effort of feature alignment.* While the performance and training time reductions are reported in Table 2 of the paper, we have also updated the Appendix (Figure 7) where we observe that bad augmentations (where regions of interest are removed) are discarded by this module.
> - **Contribution III:** Though ideas around inter-class variability have been explored in prior literature in other contexts, to the best of our knowledge, they have not been used in a multimodal model like CLIP in a few shot setting. In prior works (like class imbalanced learning), the classifier weights have been separated by regularization techniques. In our context, if we similarly train the text prompts with a regularizer such that the text embeddings uniformly separate out in the feature space, we observe that there is no consistent separation between the text and image features in the multimodal representation space. *This issue is particular for multimodal models like CLIP, where regularizing the text embeddings is not sufficient to separate out the visual embeddings in a few-shot setting.*
>   Our proposed MMReg module solves this issue by enforcing both the image and text modality features to separate out uniformly, while bringing them closer to each other in the multimodal embedding space. To illustrate this, we consider the four datasets from the ablation studies, and report the performances with prompt tuning (MaPLe), prompt tuning with only text regularizer, and training with MMReg in Table A (here). We observe that simply separating text features (classifiers) reduces performance than MaPLe, while the MMReg improves across all the cases. We have also added the interclass L2 distance heatmaps for two benchmark datasets in the Appendix (Figure 5) to illustrate this.
>
> [1] A. Radford, et. al., "Learning Transferable Visual Models From Natural Language Supervision", ICML (2021).

---

> > ### Author Response · Authors · 2024-11-06
> > **Response to Reviewer DSBP (2/2)**
> >
> > |                              | Omniglot       | Plantae        | ISIC           | Plant Disease    |
> > |------------------------------|----------------|----------------|----------------|----------------|
> > | Maple + sel. augs            | 85.49          | 64.69          | 33.54          | 82.66          |
> > | Maple + sel. augs + text reg | 85.76          | 64.65          | 33.44          | 82.54          |
> > | Maple + sel. augs + MMReg    | **87.01** | **66.13** | **33.90** | **83.48** |
> >
> > TABLE A: Effect of prompt tuning with text regularization vs MMReg.
> >
> > We would like to emphasize that the contributions of our work lie in the motivation and simplicity of the proposed modules and we feel that this work can serve as an important baseline for transfer learning of VLMs in such challenging and realistic downstream tasks.
> >
> > >While the MMReg aims to improve generalization by separating class embeddings, the model’s ability to handle highly specialized or unfamiliar domains (e.g., medical images) still relies on the presence of meaningful class names, which may not always be available.
> >
> > In Section 6.3 of the paper, we have observed that in some specialized datasets like Plantae (with scientific plant names), the proposed MMReg module improves the performance over the strong baseline MaPLe by a significant margin of +2.7 \%, even when we replaced them with placeholder classnames, with only a single example per class. The reviewer observes that since in medical domain datasets like ISIC and Chest X-Ray, meaningful classnames have been used, the model may still rely on them for proper classification in the CLIP space. To emulate such a scenario where classnames may not be available, we replace the classnames of these two specialized datasets with pseudo classnames (like C1, C2, C3, etc.), and see the effect of the MMReg module on them (Table B here). We observe from the results that while performance of MaPLe (prompt tuning) drops because the class semantics are now missing, our MMReg module helps them separate out in the multimodal representation space, thereby improving the performance in both the datasets, which conforms with our motivation for this proposed module.
> >
> > | Dataset                             | Plantae        | ISIC           | ChestX         |
> > |-------------------------------------|----------------|----------------|----------------|
> > | MaPLe (\%) (with classnames)        | -              | 31.96          | 21.30          |
> > | MaPLe (\%) (with pseudo classnames) | 55.34          | 28.32          | 20.19          |
> > | MaPLe + MMReg (\%) (with pseudo classnames)     | **58.04** | **30.24** | **20.89** |
> >
> > TABLE B: Effect of MMReg when classnames are not available.

---

### Review · Reviewer_EUMS · 2024-11-02

**Summary Of Contributions:**

The paper proposes PromptMargin to adapt large-scale VLMs for few-shot learning under distribution shifts.

- The authors provides a detailed analysis on CLIP’s embedding space to determine conditions under which it performs well on a target dataset and when fine-tuning is beneficial.

- Moreover, the prompt-learning framework is proposed to directly adapt VLMs to few-shot tasks without pre-training on a large source dataset. The method is "source-free", meaning that it relies only on target data samples.

- To improve class separation and generalization, the authors further propose a regularizer to ensure that text and image embeddings maintain a consistent separation in the embedding space.

- The empirical results are presented on 15 different datasets under various few-shot settings, showing superior performance to existing baselines in most cases.

**Audience:**

Yes

**Claims And Evidence:**

Yes

**Requested Changes:**

- Include additional qualitative or visual analyses (e.g., t-SNE) to illustrate how the MMReg affects class separability in embedding space. This would provide valuable insights into the internal mechanics of the framework and support claims about improved class discrimination.
- A detailed hyperparameter sensitivity analysis would be beneficial. For example, a brief exploration of the sensitivity of the model's performance to key hyperparameters including number of augmentations and MMReg separation margin could help further understanding the method.

**Strengths And Weaknesses:**

### Strengths

- Experiments were extensive on 15 datasets with varying distribution shifts to demonstrate the robustness of PromptMargin, which showed improved performance over existing methods in most cases.

- The selective augmentation strategy is practical, as it optimizes training time while maintaining or even improving model accuracy, which is essential for few-shot scenarios.

- The ablation studies provide insight into the importance of each module within PromptMargin, helping to justify the effectiveness of each component.

### Weaknesses
- In cases where target data is closely aligned with the source distribution (e.g., mini-ImageNet), PromptMargin sometimes underperforms compared to baseline methods.
- The proposed approach’s reliance on multiple augmentations during training may increase computational overhead in some cases

---

> ### Author Response · Authors · 2024-11-06
> **Response to Reviewer EUMS (1/2)**
>
> We thank the reviewer for acknowledging the practicality of the selective augmentation strategy and the extensive experiments performed. We now address the concerns.
> >In cases where target data is closely aligned with the source distribution (e.g., mini-ImageNet), PromptMargin sometimes underperforms compared to baseline methods.
>
> Foundation models like CLIP have been extensively trained on huge datasets. Thus for some natural image datasets like mini-ImageNet and Aircraft, the zero-shot performance of CLIP is itself very high (e.g., 99.21\% accuracy on mini-ImageNet). As the reviewer observed, for such datasets, further tuning CLIP may actually reduce the performance, as it may distort the CLIP semantic space. To address this issue, we have proposed a simple, yet effective method to estimate the domain/semantic differences of the different datasets to understand when CLIP should be finetuned for improved performance.
> Thus, in real scenarios, the proposed framework can be applied only to those datasets where CLIP is not able to generalize and perform satisfactorily due to significant domain shifts or presence of semantically unknown/meaningless classnames. We observe that for these challenging datasets, the proposed framework significantly outperforms the zero-shot performance and also the stronger baselines like Maple.
> >The proposed approach’s reliance on multiple augmentations during training may increase computational overhead in some cases
>
> We agree with the reviewer that incorporating augmentations will increase computational overhead. However, augmentations have been a common choice for many prior few-shot methods, and our proposed selective augmentation strategy reduces the training time by 40\% as well as the GPU memory by 4.88\%, which may be further reduced with efficient coding. On the other hand, alternative methods like meta-learning require much more compute than simple augmentation strategies.
> >Include additional qualitative or visual analyses (e.g., t-SNE) to illustrate how the MMReg affects class separability in embedding space. This would provide valuable insights into the internal mechanics of the framework and support claims about improved class discrimination.
>
> As suggested, we have added the t-SNE visualizations of the image embeddings after training with our proposed MMReg module in the Appendix (Figure 6) for two benchmark datasets, Omniglot and Plant Disease. We observe that training with the MMReg regularizer uniformly separates and more compactly clusters the few-shot image examples in the feature space. We have also added the interclass L2 distance heatmaps (Figure 5 in Appendix) which illustrate the uniform separation between the classes in both the textual and visual feature spaces.
> >A detailed hyperparameter sensitivity analysis would be beneficial. For example, a brief exploration of the sensitivity of the model's performance to key hyperparameters including number of augmentations and MMReg separation margin could help further understanding the method.
>
> As suggested, we provide a hyperparameter sensitivity analysis here. Table A (here) shows the effect of the number of selected augmentations. For very few augmentations (say, 5), due to less number of examples, the results are in general lower. However, we observe that the performance stabilizes for around 10 augmentations.

---

> > ### Author Response · Authors · 2024-11-06
> > **Response to Reviewer EUMS (2/2)**
> >
> > |               | EuroSAT | ISIC |
> > |---------------|--------------------------------------|-----------------------------------|
> > | Augmentations | Accuracy (\%)                        | Accuracy (\%)                     |
> > | 20 augs       | 79.39                                | 33.60                             |
> > | 5 sel. augs   | 75.53                                | 28.13                             |
> > | 10 sel. augs  | 81.47                                | 34.47                             |
> > | 15 sel. augs  | 78.26                                | 34.40                             |
> > |   |   |   |
> > | 30 augs       | 79.20                                | 33.20                             |
> > | 5 sel. augs   | 76.53                                | 27.07                             |
> > | 10 sel. augs  | 77.47                                | 36.27                             |
> > | 15 sel. augs  | 81.93                                | 35.47                             |
> > | 20 sel. augs  | 79.53                                | 33.86                             |
> > |     |   |    |
> > | 45 augs       | 78.26                                | 33.86                             |
> > | 5 sel. augs   | 79.47                                | 26.73                             |
> > | 10 sel. augs  | 79.53                                | 34.40                             |
> > | 15 sel. augs  | 78.47                                | 35.33                             |
> > | 20 sel. augs  | 79.40                                | 35.40                             |
> > |     |   |    |
> > | 60 augs       | 78.73                                | 33.26                             |
> > | 5 sel. augs   | 75.13                                | 25.86                             |
> > | 10 sel. augs  | 77.67                                | 29.80                             |
> > | 15 sel. augs  | 79.73                                | 33.66                             |
> > | 20 sel. augs  | 80.33                                | 28.73                             |
> >
> > TABLE A: Sensitivity analysis on selective augmentations for the same 20 sampled episodes.
> >
> > In Table B (here), we vary the weighting coefficients ($\alpha$ and $\beta$) of the two terms of the MMReg module $\mathcal{R}(\tilde{X_T})$ and $\mathcal{R}(\tilde{X_V})$ respectively for two benchmark datasets, EuroSAT and ISIC.
> > In our work, we use $\alpha=1$ and $\beta=1$ for all experiments, since hyperparameter tuning is not possible for few-shot learning.
> > But we observe that the performance is quite stable for a wide range of parameters.
> >
> > | Dataset        |Weighting Coefficients| Accuracy |
> > |--------------------------|:-----------------------------------------------------:|-------------------|
> > |                          | **($\alpha$ , $\beta$)**     | |
> > |  |      | |
> > |  | (1, 1)                 | 76.33 $\pm$ 0.095 |
> > | EuroSAT| (5, 10) | 77.61$\pm$ 0.019 |
> > || (10, 5)| 77.49$\pm$ 0.018  |
> > |  |                  | |
> > |  | (1, 1)                 | 32.93$\pm$0.064 |
> > | ISIC| (5, 10) | 32.77$\pm$0.062 |
> > || (10, 5)| 32.88$\pm$ 0.063 |
> >
> > TABLE B: Hyperparameter sensitivity analysis for $\alpha$ (for $\mathcal{R}(\tilde{X_T})$) and $\beta$ (for $\mathcal{R}(\tilde{X_V})$).

---

### Author Response · Authors · 2024-11-06
**Added Appendix**

We have added an Appendix to the paper which includes additional visual analyses to better explain our proposed PromptMargin framework. As suggested by Reviewer EUMS, we have included t-SNE visualizations of the image embeddings after training with our proposed MMReg module, where we observe that the MMReg regularizer uniformly separates and more compactly clusters the few-shot image examples in the feature space. We have also added the interclass L2 distance heatmaps which illustrate the uniform separation between the classes in both the textual and visual feature spaces after training with MMReg. Additionally, we have included qualitative analyses on the types of augmentations being selected by the proposed selective augmentation method, and the effect of strong vs weak augmentations, as suggested by Reviewer 8apX.

---

### Author Response · Authors · 2024-11-15
**General Response to the Reviewers**

Dear Reviewers,

We would like to thank you for your insightful reviews and valuable time. We have tried to address the concerns to the best of our ability and also revised the manuscript to include additional visual analyses as suggested. Please let us know if you have any further questions or clarifications from your end. We would be happy to address them in the comments.

Thank you.

---

### Decision · Action_Editor_kNc4 · 2024-12-09

**Recommendation:** Accept as is

**Comment:**

This paper investigates few-shot learning via prompt tuning in CLIP models where the few-shot tasks are out-of-distribution. Unlike past work that has tackled such OOD tasks with CLIP by first finetuning the model on a larger dataset before the few-shot adaptation, the method proposed in this work is "source-free" and directly performs that adaptation. They first investigate, by inspecting distances in embedding space, whether we can estimate the performance of zero-shot CLIP on a new task. They then propose a framework for improved source-free adaptation by enhancing prompt tuning with selective data augmentation as well as a regularizer designed to encourage uniform separation of image and text category embeddings.

The reviewers found that the paper is well-written and the motivation is clear (Reviewer 8apX), the experimental investigation is extensive, and the ablations studies insightful (Reviewers EUMS and 8apX). Reviewers brought up weaknesses with respect to computational overhead, limitations of the method in terms of not improving in all scenarios over baselines (improvements are seen clearly when there is a large distribution shift but not when the source and target domains are similar) (Reviewer EUMS). The authors have clarified the strengths and limitations of their work and discussed computational overhead (which is no worse than competing methods). Reviewer DSBP brought up the limitation of reliance on "meaningful class names". During the rebuttal, the authors addressed this by running additional experiments without the semantically-meaningful names. The authors also ran experiments with another model during the rebuttal, in response to Reviewer 8apX's feedback that the experimental investigation is limited to CLIP.

Based on the above, I recommend the paper to be accepted to TMLR; the reviewers found the claims backed by sufficient evidence, and the findings of interest to the community.

**Audience:**

Few-shot learning in vision-language models is of interest to the community of TMLR. While some reviewers thought that the work lacks "novelty", others found various insights of this work valuable and interesting (e.g. Reviewer 8apX's comments relating to the analysis of embedding space distances and implications for performance after finetuning being insightful). The fact that the proposed method outperforms prior work on several OOD benchmarks will also be of interest to the community.

**Claims And Evidence:**

This paper investigates few-shot learning via prompt tuning in CLIP models where the few-shot tasks are out-of-distribution. Unlike past work that has tackled such OOD tasks with CLIP by first finetuning the model on a larger dataset before the few-shot adaptation, the method proposed in this work is "source-free" and directly performs that adaptation. They first investigate, by inspecting distances in embedding space, whether we can estimate the performance of zero-shot CLIP on a new task. They then propose a framework for improved source-free adaptation by enhancing prompt tuning with selective data augmentation as well as a regularizer designed to encourage uniform separation of image and text category embeddings. They claim their method is effective compared to the state-of-the-art for few-shot learning under distribution shifts.

The reviewers found that the claims were backed by experiments that were "extensive on 15 datasets with varying distribution shifts to demonstrate the robustness of PromptMargin, which showed improved performance over existing methods in most cases" and they found that "the ablation studies provide insight into the importance of each module within PromptMargin, helping to justify the effectiveness of each component" (Reviewer EUMS). Reviewer 8apX noted that "The experiment show that the PromptMargin framework indeed improves over the baseline methods." and that "The ablation studies show that selective augmentation boosts the performance the most. This is an interesting insight."